# Reduction in the Duration of Postoperative Fever during the COVID-19 Pandemic in Orthopedic and Traumatic Surgery Due to PPE and Cautions

**DOI:** 10.3390/jcm11061635

**Published:** 2022-03-16

**Authors:** Pierluigi Mastri, Francesco Di Petrillo, Alessandro Cerone, Mario Muselli, Michela Saracco, Giandomenico Logroscino, Vittorio Calvisi

**Affiliations:** 1Department of Life, Health & Environmental Sciences, University of L’Aquila, 67100 L’Aquila, Italy; dipest@hotmail.it (F.D.P.); cerone.ale@gmail.com (A.C.); mario.muselli@gmail.com (M.M.); giandomenico.logroscino@univaq.it (G.L.); vittorio.calvisi@univaq.it (V.C.); 2Department of Orthopaedics, Fondazione Policlinico Universitario A. Gemelli IRCCS, Università Cattolica del Sacro Cuore, 00168 Rome, Italy; michelasaracco@gmail.com

**Keywords:** postoperative fever, fractured patients, PPE, COVID-19, SARS-CoV-2

## Abstract

The Italian government on the 8th of march in response to the increased global prevalenceof severe acute respiratory syndrome coronavirus 2 (SARS-CoV-2) stated a national quarantine. In this period the absence of rapid and sure screening tests for COVID-19 made necessary more attention for presence of fever in hospitalized patients, like continuous use of surgical, FFP2, and FFP3 masks (PPE) by nurses, physicians, and patients; moreover, patients visits were restricted. In this period, during the daily activities in our orthopedic department we observed an empirical decreased incidence of post-operative fever in patients admitted for trauma surgery. The aim of this study is to compare the prevalence of post-operative fever in this period with the same period in 2019. We analyzed the presence of post-operative fever in 110 patients admitted in 2020 and 129 admitted in 2019. The results show a significant decrease of the prevalence and duration of post-operative fever in patients admitted in 2020. This study evidenced that the use of PPE and limitation in external access to the hospital decrease postoperative fever in hospitalized patients undergone surgery for fracture.

## 1. Introduction

Postoperative fever is a common complication following orthopedic trauma surgery, with an incidence between 10% and 40% depending on definitions [1,2]. The causes of the febrile response can be many, including acute infection and general inflammatory response. The Italian government on the 8 March 2020 in response to the increased global prevalence of severe acute respiratory syndrome coronavirus 2 (SARS-CoV-2) stated a national quarantine [3,4]. At the beginning of the infection, to prevent COVID-19 disease in nurses, doctors, and patients, there were no screening tests and prevention tools such as surgical masks, FFP2, and FFP3 (PPE) were introduced in the daily clinical practice. In addition, visits by relatives to sick patients were limited or forbidden. During this period in our Department of Orthopedics in “Ospedale Regionale S. Salvatore” of L’Aquila we observed an empirical decreased incidence of post-operative fever in patients admitted for trauma surgery.

As the duration of the surgical operation, the protocols of antibiotic prophylaxis and the other causes related to postoperative fever were the same, we hypothesize that this lower incidence was related to the concomitant increased precautions used during hospitalization in the early period of SARS-CoV-2 infection, such as the use of PPE (Personal Protective Equipment) and no external visits to patients.

The aim of this study is to compare the incidence of post-operative fever in the first wave of the pandemic period (March/June 2020) with the same period in 2019, when there was no SARS-CoV-2 infection.

## 2. Materials and Methods

We analyzed 129 patients admitted in the period from 8 March 2020 to 3 June 2020. All the patients were hospitalized for trauma surgery, 46 patients with severe trauma involving the lower limb, 8 for severe trauma involving the upper limb, 4 polytrauma, 52 for minor trauma involving the upper and lower limbs. During this period, we have discontinued elective surgery. We excluded 2 open fractures, 6 revisions in hip periprosthetic infection, 8 implant removal, and 3 curettage of wound because these cases could confound the prevalence of postoperative fever. Patient temperatures were measured twice a day. Finally, we selected 110 records of patients hospitalized during the pandemic period (from 8 March 2020 to 3 June 2020) that were compared with those of the same period (8 March 2019 to 3 June 2019) in the absence of SARS-CoV-2 infection.

In the non-pandemic period (8 March 2019 to 3 June 2019), 241 patients were screened. We excluded 100 patients admitted for elective surgeries and 4 revisions in hip periprosthetic infection, 6 implant removal and 2 curettage of wound; no open fractures were hospitalized during this period. A total of 129 records were included in this group 51 for major trauma involving the lower limb, 13 for major trauma involving upper limb, 54 for minor trauma involving upper and lower limb, and 11 for different procedures.

Postoperative fever was considered with a cut off over 37.5 °C in a timeframe from the day after surgery to a maximum of 14 days after surgery; we recorded also the duration of post-operative fever during hospitalization, time of surgery, and hospitalization length. All the data were than analyzed statistically.

### Statistical Analysis

Statistical analyses was performed using Stata (StataCorp. 2015. Stata Statistical Software: Release 14. College Station, TX, USA: StataCorp LP.) After ascertaining the skewed distribution of all data with the Shapiro–Wilk test, the Wilcoxon rank-sum test was used to compare continuous variables, while proportional differences were assessed using the chi-square test. A *p* value of <0.05 was considered significant.

## 3. Results

A total of 239 patients data were collected; there were 127 (53.1%) males and 112 (46.9%) females, with an average age of 63.2 ± 22.3 years (range, 4 to 99). The age of males was 70.6 ± 19.2 years, significantly older than that (54.7 ± 22.6) of females (z = 5600, *p* < 0.0001). Comparing the two groups (2020 vs. 2019), the number of admissions declined from 129 in 2019 to 110 in 2020 (14.7%) (Table 1). The different gender distribution in the years considered was not significant: In 2019, there were 66 (51.2%) males and 63 (48.8%) females; in 2020, there were 61 (55.5%) males and 49 (44.5%) females.

The age of 2020 group was 64.7 ± 19.1 years; the age of 2019 group was 61.9 ± 24.6 years: such difference was not statistically significant. The average duration of the surgical procedure was not significantly different between 2019 and 2020 (Table 1). Similarly, there were no statistically significant differences in the average duration of hospitalization (*p* = 0.8040). During 2020, less post-operative fever was recorded than in 2019: 24 of the 2020 patients (21.8%) and 76 of the 2019 patients (58.9%), *p* < 0.0001). In the 2020 group, fever was observed in the first post-op period in 13.6% (15) patients, while after 4 days, fever was observed in 8.2% (9) patients. In the 2019 group, 35.6% (46) of the fever cases were in the first post-op period, while 23.3% (30) were after 4 days. In addition, the average duration of post-operative fever in 2020 was significantly shorter than in 2019 by 1.4 days (*p* = 0.0004) (Figure 1).

Regarding the onset of fever, in the 2020 group we observed that 62.5% (15) of post-op fever appeared in the first 4 days after surgery and 37.5% (9) 4 days after surgery (8.2% of the total number of hospitalized patients; *N* = 110). Totally, 7 had infective complications like pneumonia, cystitis, or surgical wound infections (6.3% of the total number of hospitalized patients (*N* = 110), 29% of total post-op fever (*N* = 24), and 77.8% of the total 4 days post-op fever patients (*N* = 9).

In the 2019 group, we observed that 60.52% (46) of post-op fever appeared in the first 4 days after surgery and 39.4% (30) 4 days after surgery (23.3% of the total number of hospitalized patients; *N* = 129). Totally, 11 had infective complications like pneumonia, cystitis, or surgical wound infections (8.5% of the total number of hospitalized patients (*N* = 129), 14.4% of total post-op fever (*N* = 76), and 36.7% of the total 4 days post-op fever patients (*N* = 30).

Therefore, during 2020 we observed a reduction in global post-op fever (21.8% vs. 58.9%), a reduction in post-op fever 4 days after surgery: from 23.3% of patients in 2019 to 8.2% in 2020 (*p* = 0.002). However, the prevalence of infective complications remained unchanged over the two years (8.5% in 2019 vs. 6.3% in 2020; *p* = 0.528). Considering the infective complications occurred in patients who had post-operative fever after 4 days of surgery, the proportion increases statistically significantly in 2020 (36.7% in 2019 vs. 77.8% in 2020; *p* = 0.030) (Figure 2).

## 4. Discussion

This study shows a significant decrease of the prevalence (21.8% vs. 58.9%; *p* < 0.0001) and duration (1.8 days vs. 3.2 days; *p* < 0.0004) of post-operative fever in patients admitted in 2020, compared to those admitted in the same period in 2019 (Figure 1); furthermore, during 2020 we observed a reduction in post-op fever 4 days after surgery while the prevalence of infective complications remained unchanged over the two years, considering the infective complications occurred in patients who had post-operative fever after 4 days of surgery, the proportion increases statistically significantly in 2020 (Figure 2). Although it is not statistically significant, there is a reduction in the length of hospital stay and this results in saving the costs of health care, and improved patient health management.

The two groups of patients (2019 and 2020) were comparable for gender, age, type of fracture. Consequently, the significant decrease in the prevalence and duration of the fever is mainly imputable to the contagion containment measures taken for the COVID-19 pandemic, in particular, the use PPE by nurses, physician, and patients, and the restricted access of the relatives to the department. Early postoperative fever is a common event and is infrequently associated with infection [5]. Literature suggests that any kind of workup in the absence of localizing symptoms in the third post-operative day or before is unwarranted and is an inappropriate use of hospital resources [6].

Indeed fever, even up to the seventh postoperative day, is not substantially helpful to distinguish infection from general inflammation in clean orthopedic surgery [7]. Under the third post-operative day, patients who experienced multiple febrile episodes, or temperatures over 38.5 °C or who underwent revision surgery are at the greatest risk of having an underlying infection [6].

In the later postoperative period, physicians should be more suspicious for an infective source of fever because a traumatic inflammatory etiology of fever is less likely [8]. Postoperative fever can have many etiologies. Andres et al. [9] evaluated serum and drain concentrations of interleukin 1β, interleukin 6 (IL-6), and tumor necrosis factor. Patients with postoperative fevers had substantially higher drain and serum concentrations of IL-6. They determined that postoperative fevers can at least partly result from the release of endogenous pyrogens as part of the inflammatory response [5,9].

The literature shows that postoperative fever is a common occurrence after surgery for orthopedic trauma, but there are no single prevalence data in the studies [8]. Post-operative fever shows an incidence ranging from 18% to 60%; the range is so high [6,7,8] because many do not care about a fever in the first few days as it is not related to a clear sign of infection and so many do not report it. We observed a prevalence of 58.9% in 2019 and 21.8% in 2020. This result is highly significant. The aim of our study is not to evaluate the prevalence of post-operative fever, but to observe, as far as possible, in the same environmental conditions, the same ward, the same healthcare personnel, the same clinical conditions, age and type of trauma, the variation of this prevalence due to the use of PPE and cautions due to the COVID-19 pandemic, which have created a particular new condition.

Indoor air quality in hospitals has been specifically considered in terms of its impact on health. Air quality is an important risk factor influencing the health of staff and patients who are in contact with indoor air inhaled in hospitals [10].

Hospital personnel and visitors are in direct contact with various types of pollutants in medical environments. Inhalation is the most common type of contact that is often neglected by the general public.

Jung et al. showed that checking real-time air pollution offers a lot of information about the effects of activities on indoor air pollution in hospitals [11].

Pollutants include pathogenic microorganisms and airborne bio-aerosols such as bacteria and spores of fungi that enter the body through inhalation and ingestion. These biological pollutants can intensify the health hazards of personnel and violate the process of recovery from infectious, respiratory, digestive, and neuropathic problems. Furthermore, the presence of more people in the same ambient causes a significant increase of biological pollutants in indoor air.

These data agree with our results, as we observed a reduction in the prevalence of post-operative fever correlated with a decrease in the number of people in the ward.

Although the causes of postoperative fever are unknown and even if it is not necessarily related to a postoperative infection, our study suggests that postoperative fever appears to have a correlation with biological contaminants. 

Moreover, this study highlighted how the prevention of air infection, by PPE use and restricted access, permit to identify more clearly the most serious and infective complicated cases of post-op fever. Indeed, this study showed that of the 4 days post-op fever, 77.8% of the 2020 PPE-protected patients had infective complications compared to 36.7% of 2019 ones. This suggest that under prevention of air infection by PPE, evidence of a 4-days post-op fever is of major concern for complicated infection risk and justifies more deep instrumental investigations and therapeutic measures. On the contrary, in the normal clinical practice, as in the cases of 2019 patients, not PPE protected, adjunctive investigations and therapies are justified only in 36.7% of the patients. In all the other cases (64.3% of fevers) these may be unnecessary. The decrease of diagnostic tests such as chest x-ray and less use of antibiotic empirical therapy, lead to a reduction in costs of health care, with a relative significant spare of resources and money.

In addition, a decrease in the prevalence of post-operative fever allows to identify earlier and more specifically the cases at real risk of post-operative infective complication such as pneumonia or wound infection.

The limitations of this study are the small number of patients enrolled and the absence of a controlled randomized group. Moreover, there was not a comparative analysis between the two groups with regard to the severity and the mechanism of the trauma (i.e., Injury Severity Score or other scores).

## 5. Conclusions

In the very first pandemic period, the absence of rapid and sure screening tests for COVID-19 made necessary more attention for presence of fever in hospitalized patients [12,13,14]. This led to stronger measures to prevent respiratory infection (PPE and restricted access to the patients), with a significative decrease of postoperative fever and hospitalization length. These results are very interesting considering that these simple measures can lead to better clinical management of the patient and to economic savings, limiting inappropriate therapies and diagnostics. Moreover, the prevention of respiratory transmitted inflammation responsible of fever allows to identify fever related with complication like early infection or other complication. This study evidenced that the use of PPE and limitation in external access to the hospital decrease postoperative fever in hospitalized patients undergone surgery for fracture. It is desirable in the future, after the pandemic is over, that more large, randomized control trials studies could confirm these results.

## Figures and Tables

**Figure 1 jcm-11-01635-f001:**
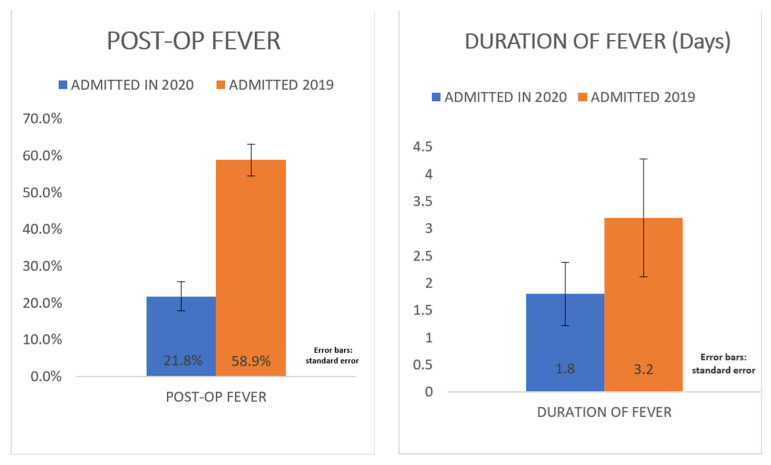
Difference between presence of post-op fever and post-op duration of fever in 2019 and in 2020.

**Figure 2 jcm-11-01635-f002:**
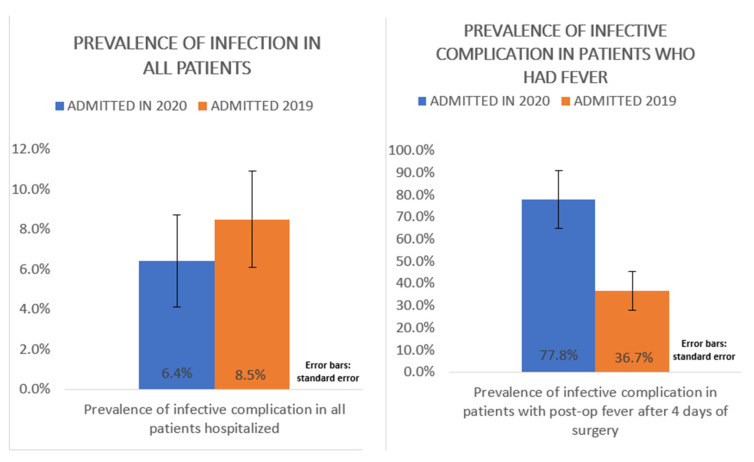
Difference between prevalence of infective complications in all patients hospitalized and in patients who had post-op fever after 4 days of surgery.

**Table 1 jcm-11-01635-t001:** Results table and comparison between the two groups.

		2019	2020	*p*
Age (years)		61.9 ± 24.6	64.7 ± 19.1	0.7439 ^1^
Gender	M	66 (51.2%)	61 (55.5%)	0.508 ^2^
	F	63 (48.8%)	49 (51.2%)	
Duration of the surgical procedure (minutes)		98.3 ± 70.3	90.4 ± 61.4	0.2805 ^1^
Length of hospitalization (days)		10.9 ± 8.7	10.1 ± 7.4	0.8040 ^1^
Post-surgical operation fever		76 (58.9%)	24 (21.8%)	**<0.0001** ^2^
Length of fever (days)		3.2 ± 2.2	1.8 ± 1.2	**0.0004** ^1^
Post-op fever 4 days after surgery		23.3%	8.2%	**0.002** ^2^
Prevalence of infective complications		8.5%	6.4%	0.528 ^2^
Prevalence of infective complications in post-op fever after 4 days of surgery		36.7%	77.8%	**0.030** ^2^

Statistically significant data in bold (*p* < 0.05). ^1^ Wilcoxon rank-sum test; ^2^ Chi-square test.

## Data Availability

The data presented in this study are available on request from the corresponding author.

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
