# Peer review of "Reduction in the Duration of Postoperative Fever during the COVID-19 Pandemic in Orthopedic and Traumatic Surgery Due to PPE and Cautions"

_jcm, 2022, doi:10.3390/jcm11061635_

Round 1
Reviewer 1 Report
Thank you for the opportunity to review this interesting manuscript, in which the authors report that postoperative fever is decreased after fracture surgery during a time of increased PPE use due to COVID-19.
- What timeframe was considered postoperative fever? Was there a cut-off for the day postop?
- Do the authors have data on the time from surgery until onset on fever? As the authors point out in their discussion, fevers immediately following surgery are not often associated with infection, while later-onset fevers with greater duration are more likely due to infection. It would be interesting to see if in this study, the later fevers are the ones that have the greatest reduction from 2019 to 2020, and this would strongly support the idea that increased precautions are reducing postop fevers by limiting bacterial infections.
- Were there any exclusion criteria for the 2020 period?
- Could the mechanism of injury be markedly different between the 2 time periods due to travel restrictions etc. E.g. fewer road traumas in 2020? Could this produce less severe injuries in the 2020 group? While broad injury category is reported, is injury severity score or similar available to properly compare the 2 groups? If not, this should be discussed as a limitation.
- Were open fractures included in this study? Would open fractures be more susceptible to postop bacterial infections? Was there a difference in the number of open fractures between the 2019 and 2020 cohorts?
- I feel like too much of the discussion is devoted to air quality when this is only one of many factors to consider.
- Another discussion point may be the reduced spread of pathogens among patients via staff as a result of increased PPE use and hand sanitisation.
- Other limitations that should be discussed are that this study did not examine postop infection rates, which would provide more definitive proof.
Author Response
Dear reviewer, we thank you for your suggests and we appreciate very much your remarks that surely will improve the quality of the study.
- What timeframe was considered postoperative fever? Was there a cut-off for the day postop?
We considered a timeframe of postoperative fever a Temperature > 37.5 °C rised up from the day after surgery (early morning measurement) to a maximum of 14 days after surgery. (Added in the text pg. 2 line 69).
- Was there a cut-off for the day postop?
Regarding the time from surgery until onset of fever we collected the data from first day post-op to the dismission of the patients with a cut-off of the 1st and the 4th day.
- Do the authors have data on the time from surgery until onset on fever?
No. We did not collected this data. We just distinguished between early fevers (in the first 3 days post-op) and the fevers after the 4th day.
- As the authors point out in their discussion, fevers immediately following surgery are not often associated with infection, while later-onset fevers with greater duration are more likely due to infection. It would be interesting to see if in this study, the later fevers are the ones that have the greatest reduction from 2019 to 2020, and this would strongly support the idea that increased precautions are reducing postop fevers by limiting bacterial infections.
Yes. We observed a clear reduction in later fevers (more than 4 day) from 23,3% in 2019 to 8,2% in 2020. During 2020 we observed a reduction in global post-op fever (21,8% vs 58,9%). (Added in the text pg.3 line 94).
- Were there any exclusion criteria for the 2020 period?
Yes. We have added exclusion criteria in M&M. We excluded open fractures, hip revisions in periprosthetic infection, implant removal and curettage of wound. (Added in the text pg.2 line 54 and 62).
- Could the mechanism of injury be markedly different between the 2 time periods due to travel restrictions etc. E.g. fewer road traumas in 2020? Could this produce less severe injuries in the 2020 group?
- While broad injury category is reported, is injury severity score or similar available to properly compare the 2 groups? If not, this should be discussed as a limitation.
Surely we observed a reduction of road trauma in 2020, and this is confirmed by the reduction of the patients (110vs129) but we didn’t analyzed the mechanism and the severity of the trauma. We add this point in the limitation of the study. (Added in the text pg.2 line 54 and 62).
- Were open fractures included in this study? Would open fractures be more susceptible to postop bacterial infections? Was there a difference in the number of open fractures between the 2019 and 2020 cohorts?
Open fractures were excluded from the study as specified in the exclusion criteria. (Added in M&M in the text pg.2 line 54 and 62).
- I feel like too much of the discussion is devoted to air quality when this is only one of many factors to consider.
We reduced that part of the discussion.
- Another discussion point may be the reduced spread of pathogens among patients via staff as a result of increased PPE use and hand sanitisation
We described and discussed in the study that effectively, with the use of PPE by the staff (patients, physician, nurses and sanitary personnel ) and with restricted admisson of outside people to the hospital, the prevention of air transmitted infection was achieved, as demonstrated by the significant decrease of postop fever.
- Other limitations that should be discussed are that this study did not examine postop infection rates, which would provide more definitive proof.
We examined and added the post infection rate or infective complication rate (pneumonia, cystitis, wound infection): in the two group it was similar (6.3% Vs 8,5%). This explain how the PPE and people restriction access are effective in preventing air or contact infections that do not evolve in serious complication fevers for which it is not necessary to add instrumental investigations and prolonged therapies; unfortunately these measures are ineffective on other more serious infection (surgical wound, cystitis, etc.) for which high suspicion and strong diagnostic therapeutic actions have to be started especially in case of more than 4 days fever.

Reviewer 2 Report
The authors compared the difference between presence of ostop fever and postop duration of fever in 2019 and in 2020. The data is scientific, but the significance did not meet the level of journal. Only two tables and one figure is presented with limited information.
Author Response
Dear reviewer we have read your review and we improved M&M section, we improved the table with other data collected to strongly increase the statistical significance to the extent possible. Furthermore we improved tables adding two new graphs. Thanks again for your suggests and your time.